# FABP7 Facilitates Uptake of Docosahexaenoic Acid in Glioblastoma Neural Stem-like Cells

**DOI:** 10.3390/nu13082664

**Published:** 2021-07-30

**Authors:** Won-Shik Choi, Xia Xu, Susan Goruk, Yixiong Wang, Samir Patel, Michael Chow, Catherine J. Field, Roseline Godbout

**Affiliations:** 1Department of Oncology, Cross Cancer Institute, University of Alberta, Edmonton, AB T6G 1Z2, Canada; wonshik@ualberta.ca (W.-S.C.); xia5@ualberta.ca (X.X.); yixiong@ualberta.ca (Y.W.); Samir.Patel2@ahs.ca (S.P.); 2Department of Agricultural, Food and Nutritional Science, University of Alberta, Edmonton, AB T6G 2E1, Canada; sgoruk@ualberta.ca (S.G.); cjfield@ualberta.ca (C.J.F.); 3Department of Surgery, University of Alberta, Edmonton, AB T6G 2B7, Canada; mmchow@ualberta.ca

**Keywords:** glioblastoma, docosahexaenoic acid, neural stem-like cells, B-FABP, FABP7, fatty acids, phospholipids, neurospheres

## Abstract

Glioblastoma (GBM) is an aggressive tumor with a dismal prognosis. Neural stem-like cells contribute to GBM’s poor prognosis by driving drug resistance and maintaining cellular heterogeneity. GBM neural stem-like cells express high levels of brain fatty acid-binding protein (FABP7), which binds to polyunsaturated fatty acids (PUFAs) ω-6 arachidonic acid (AA) and ω-3 docosahexaenoic acid (DHA). Similar to brain, GBM tissue is enriched in AA and DHA. However, DHA levels are considerably lower in GBM tissue compared to adult brain. Therefore, it is possible that increasing DHA content in GBM, particularly in neural stem-like cells, might have therapeutic value. Here, we examine the fatty acid composition of patient-derived GBM neural stem-like cells grown as neurosphere cultures. We also investigate the effect of AA and DHA treatment on the fatty acid profiles of GBM neural stem-like cells with or without FABP7 knockdown. We show that DHA treatment increases DHA levels and the DHA:AA ratio in GBM neural stem-like cells, with FABP7 facilitating the DHA uptake. We also found that an increased uptake of DHA inhibits the migration of GBM neural stem-like cells. Our results suggest that increasing DHA content in the GBM microenvironment may reduce the migration/infiltration of FABP7-expressing neural stem-like cancer cells.

## 1. Introduction

Glioblastoma (grade IV astrocytoma) is the most common primary brain cancer in adults [1,2]. It is a highly aggressive and deadly cancer, with a median survival time of ~15 months despite multimodality therapy [3,4]. Resistance to treatment is due in part to the invasion of surrounding brain parenchyma by GBM cells, suggesting that targeting the infiltrative properties of GBM cells might be an effective therapeutic strategy [5]. In addition to their infiltrative nature, GBMs are highly heterogeneous tumors at both the cellular and molecular levels, adding an additional level of complexity to the effective targeting of these tumors [6,7]. Multi-lineage differentiation from GBM neural stem-like cells forms the basis of GBM heterogeneity [8,9], with GBM neural stem-like cells displaying resistance to radiation and chemotherapy, thereby repopulating the tumor [10,11]. GBM neural stem-like cells exhibit distinct metabolic profiles compared to non-stem cell populations, with reduced glycolysis and higher lipid metabolism [12,13,14]. Dysregulation of lipid metabolism has also been associated with maintenance of GBM stemness and poor survival [14,15]. Important mediators of lipid metabolism, such as brain fatty acid-binding protein (B-FABP or FABP7) [16], fatty acid transporter CD36 [17], acyl-CoA-binding protein (ACBP) [18], and lipid elongation enzyme ELOVL2 [19] have all been reported to be highly expressed in GBM neural stem-like cells, highlighting the importance of lipid metabolism.

The human adult brain is highly enriched in lipids, especially long chain polyunsaturated fatty acids (PUFAs) [20,21]. The two main types of PUFAs in the brain are ω-6 arachidonic acid (AA) and ω-3 docosahexaenoic acid (DHA) [20]. DHA and AA are believed to have opposing roles in cancer [22]. AA is the precursor of ω-6 series eicosanoids such as prostaglandin E2 (PGE2), which stimulate inflammation, cancer growth, and invasion [23,24,25]. In contrast, DHA is the precursor of neuroprotectins and resolvins, which resolve inflammation and inhibit cancer growth [26,27,28]. In healthy adult brain, there is a tightly controlled balance between ω-6 and ω-3 PUFAs, with DHA levels exceeding AA levels in both total lipids and total phospholipids [29]. In human GBM tumor tissue, AA levels remain similar to that of healthy brain tissue; however, DHA levels are decreased by 50%, resulting in a significantly lower DHA:AA ratio [30,31]. DHA supplementation in GBM cells inhibits cell proliferation [32,33] and migration [34,35], and induces apoptosis [32,35,36]. Furthermore, DHA sensitizes GBM cells to ionizing radiation [37], suggesting that DHA supplementation may benefit GBM patients.

Brain fatty acid-binding protein (B-FABP or FABP7) is normally expressed in neural stem cells during development [38,39], and is also expressed in GBM stem-like cells [16,40]. FABP7, whose preferred ligands are PUFAs [41], is preferentially found at the infiltrative edges of GBM tumors [16,42] and its expression correlates with increased GBM cell migration [43]. DHA treatment has previously been shown to increase DHA content in the total lipids of U87 GBM cells [32]. Furthermore, C6 glioma cells injected directly into rat brain have elevated ω-3 PUFA levels and an increased ω-3:ω-6 ratio when rats are fed a DHA-rich diet compared to control diet [44]. Based on our previous work, DHA inhibits GBM cell migration in a FABP7-dependent manner [34]. However, these results were obtained with well-established cell lines believed to represent the more ‘differentiated’ aspect of GBM tumors. As resistance to treatment and poor prognosis has been linked to the neural stem-like tumor cell populations (a.k.a. brain tumor-initiating cells) in GBM tumors [10,11,45,46], it is important to know whether DHA treatment will have similar effects on GBM neural stem-like cells. 

Here, we examine the fatty acid composition of patient-derived GBM cells cultured in either regular medium (contains fetal calf serum and promotes a generally more differentiated phenotype) or neurosphere medium (contains B-27 supplement and growth factors and promotes a more neural stem cell-like phenotype) [47,48]. We compare the effects of DHA treatment and FABP7 expression on DHA uptake and cell migration in these two types of GBM cells. We also investigate the role of FABP7, which is highly expressed in GBM stem-like cells [40], in the uptake of DHA in lipids and phospholipids. Our results indicate that DHA treatment increases the DHA content in total lipids and phospholipids of GBM neural stem-like cells, with FABP7 expression facilitating DHA uptake. We also report a correlation between increased DHA uptake and decreased cell migration in GBM neural stem-like cells. 

## 2. Materials and Methods

### 2.1. Primary GBM Neurosphere Cultures

GBM primary cultures (A4-004, A4-007, A4-011, and A4-012) were prepared by enzymatic digestion of GBM biopsies obtained from patients who were consented prior to surgery under Health Research Ethics Board of Alberta Cancer Committee approval (HREBA.CC-14-0070). GBM cells were plated in either DMEM supplemented with 10% fetal calf serum (FCS) (for the generation of adherent cultures with a more differentiated phenotype) or DMEM/F12 medium supplemented with 0.5X B-27 (Life Technologies, Carlsbad, CA, USA), 20 ng/mL epidermal growth factor (EGF), and 10 ng/mL fibroblast growth factor (FGF) (for the generation of neurosphere cultures with a neural stem cell-like phenotype). ED501 neurosphere cultures and U251 GBM cells have been previously described [49].

### 2.2. Establishment of Stable FABP7-Depleted Cell Lines

Lentivirus shRNA packaging plasmids and control plasmids were purchased from Sigma. The two lentivirus FABP7 shRNA constructs used for our experiments were obtained from the University of Alberta RNAi Core Facility. FABP7 shRNA constructs sequences and virus production have been described previously [49]. The MISSION pLKO.1 plasmid (SHC002; Sigma-Aldrich, St. Louis, MO, USA) served as the control vector. A4-004N, U251 and ED501 cells were infected with lentivirus overnight and the medium changed. Infected GBM cells were selected in 1 μg/mL puromycin.

### 2.3. Western Blotting

For western blot analysis, we loaded 40 µg of whole cell lysates per lane. Proteins from whole cell lysates were separated by SDS-polyacrylamide gel electrophoresis and transferred to nitrocellulose membranes. Membranes were then immunoblotted with rabbit polyclonal anti-FABP7 (prepared in-house; 1:1000) [49] and mouse anti-GAPDH (1:1000; Thermo Fisher Scientific, Waltham, MA, USA) antibodies, followed by horseradish peroxidase-conjugated secondary antibodies (1:50,000; Invitrogen, Thermo Fisher Scientific, Waltham, MA, USA) using ECL Western Detection Reagent (GE Healthcare Life Sciences, Chicago, IL, USA).

### 2.4. Fatty Acid Preparation and Treatment

Fatty acids (DHA and AA) (Sigma-Aldrich, St. Louis, MO, USA) were dissolved in ethanol, then complexed to BSA (Roche) over a steady stream of nitrogen gas and stored at −80 °C under reducing conditions. Both GBM neurosphere and GBM adherent cells were cultured at 37 °C in a humidified 5% CO_2_ atmosphere. For fatty acid treatment, A4-004 neurosphere and adherent cells at 60–70% confluency were cultured in medium (neurosphere medium for A4-004N and serum-free DMEM for A4-004Adh) with 30 μM BSA (vehicle control), 30 μM DHA, or 30 μM AA for 24 h. Cells were then collected for lipid extraction and fatty acid analysis. Experiments were repeated three times. 

### 2.5. Lipid Extraction and Gas Chromatography

Total lipids were extracted with chloroform/methanol (2:1 vol/vol) using a modification of the Folch procedure [50]. Thin layer chromatography on silica G plates was used to isolate total phospholipids. Samples were scraped and methylated at 110 °C with a mixture (1:1) of BF3/methanol reagent (Sigma) and hexane. Fatty acid methyl esters were separated and identified by gas liquid chromatography (Agilent Model7890A, Agilent Technologies) using a 100 m CP-Sil 88 fused capillary column (100 m × 0.25 mm, Agilent Technologies, Santa Clara, CA, USA) and STD 502 (NuChek, Elysian, MN, USA) [51]. Fatty acids with a 14 to 24 carbon chain length were quantified and are presented as % of total fatty acids.

### 2.6. Semi-Quantitative RT-PCR

RNA was purified from paired patient-derived GBM neurosphere (A4-004N, A4-007N, A4-011N and A4-012N) and adherent cells (A4-004Adh, A4-007Adh, A4-011Adh and A4-012Adh) using the RNeasy Plus Kit (Qiagen, Hilden, Germany), and cDNA was generated using Superscript II reverse transcriptase (Life Technologies). The following primers were used for semi-quantitative RT-PCR analysis: FABP7 (Forward 5′-TGGAGGCTTTCTGTGCTAC-3′; Reverse 5′-TAGGATAGCACTGAGACTTG-3′), SOX-2 (Forward 5′-ACACTGCCCCTCTCACACA-3′; Reverse 5′-CATTTTTTTCGTCGCTTGGAG-3′), Nestin (Forward 5′-GGAGAAACAGGGCCTACAG-3′; Reverse 5′-GCAGAGAGAGAGGAGCATC-3′) and β-actin (Forward 5′-CTGGCACCACACCTTCTAC-3′; Reverse 5′-CATACTCCTGCTTGCTGATC-3′).

### 2.7. Lipid Droplet Analysis

GBM neural stem-like cells (A4-004N and ED501) were cultured on coverslips in neurosphere medium followed by treatment with BSA (vehicle control) or 30 µM DHA for 24 h. U251 cells were cultured in serum-free DMEM supplemented with BSA or 60 μM DHA for 24 h. Cells were fixed with 4% paraformaldehyde for 5 min at room temperature and stained with 1 µg/mL Nile Red for 15 min. Coverslips were mounted with Mowiol 4-88 mounting medium containing DAPI and images were acquired using a Zeiss confocal microscope. ImageJ software was used for quantitative analysis. We used particle analysis to identify the number of nuclei (indicative of number of cells) in each image. The images were thresholded so that particles >20 µm^2^ were identified as nuclei. The Nile Red channel was thresholded to minimize background signal and identify regions of interest (ROIs). Nile Red total intensity was calculated based on the number of pixels under each intensity (0–255) in the ROIs of each image. The average Nile Red intensity per cell was calculated based on total Nile Red intensity divided by the number of cells in each image.

### 2.8. Transwell Assay

Directional cell migration was measured using the Transwell cell migration assay. Fifty thousand cells in neurosphere medium were seeded in the top chambers of 24-well cell culture Transwell inserts (Falcon Cell Culture Inserts, Corning, Glendale, AZ, USA). Cells were allowed to migrate through an 8-μm polyethylene terephthalate (PET) membrane towards a chemoattractant (DMEM + 10% fetal calf serum) in the bottom chamber for 20 h. Fatty acids were added to culture medium for 24 h before carrying out the migration assay. Cell fixation, immunostaining and cell counts have been previously described [52]. Three independent experiments were carried out for each cell line tested.

### 2.9. Statistical Analysis 

Assessment of the significance of differences between groups was by one-way ANOVA followed by post-hoc Tukey’s test (three groups comparison) and student’s unpaired *t*-test (two groups comparison). Microsoft Excel (Microsoft, Redmond, WA, USA) and Prism 8 (GraphPad Software, San Diego, CA, USA) were used for statistical analysis of data. A *p*-value < 0.05 was considered statistically significant.

## 3. Results

### 3.1. Fatty Acid Profiles of Total Lipids Extracted from GBM Neural Stem-like versus Adherent Cells

GBM tumor tissues have elevated levels of proteins involved in fatty acid metabolism compared to normal brain [53]. Many of these proteins (FABP7, FASN and ELOVL2) are up-regulated in GBM patient-derived tumor stem cell cultures compared to more-differentiated adherent cells cultured in serum-containing medium [13,19,40]. Expression of PUFA metabolism genes in GBM neural stem-like cells suggests that it may be possible to alter the PUFA content of these cells through manipulation of the lipid microenvironment. As our previous results indicate that an increased DHA:AA ratio inhibits GBM cell migration [34], we compared the fatty acid profiles of GBM neural stem-like cells versus GBM adherent cells, and the effect of DHA and AA supplementation on their respective fatty acid profiles. 

We established GBM cultures from patients using “neurosphere” medium (promotes growth of neural stem-like cells) and “FCS-containing” medium (promotes growth of GBM cells with a more differentiated adherent phenotype) [48]. We first examined the expression of FABP7, an established GBM neural stem-like cell marker [40], in our paired GBM cultures using semi-quantitative RT-PCR. While all eight cultures expressed FABP7, levels were generally higher in the neural stem-like cell cultures compared to adherent cultures (Figure 1). Similar trends were observed when we examined the expression of the well characterized GBM neural stem cell markers, Nestin and SOX-2 (Figure 1). These results are consistent with a previous report showing higher expression of FABP7 in neurosphere cultures compared to adherent cultures, and correlation of FABP7 with Nestin and SOX-2 [40]. As the biggest difference in FABP7 levels was observed in A4-004, we used A4-004N and A4-004Adh cultures for our fatty acid profiling study. 

Total lipids were extracted from A4-004N and A4-004Adh cells cultured in their respective media and fatty acid composition was analysed by gas chromatography (Appendix A). Differences in fatty acid composition were observed for all classes of fatty acids (saturated fatty acids (SFAs), monounsaturated fatty acids (MUFAs), and polyunsaturated ω-6 and ω-3 PUFAs)) (Appendix A and Figure 2A). The most prevalent ω-3 PUFA in A4-004Adh was DHA (1% of total fatty acids), while DHA was virtually undetectable in A4-004N (Figure 2C). In A4-004N, the most abundant ω-3 PUFA was C18:3 (alpha-linolenic acid (ALA)) (1.79% of total fatty acids) (Appendix A). The most prevalent forms of ω-6 PUFA were C20:2ω-6 (3.25% of total fatty acids in A4-004N vs. 0.95% in A4-004Adh) and C20:4ω-6 (AA) (1.01% in A4-004N vs. 3.01% in A4-004Adh) (Figure 2D). The DHA:AA ratio was much lower in A4-004N compared to A4-004Adh cells (0.14 vs. 0.84), whereas the ω-3:ω-6 ratio showed no difference in A4-004N compared to A4-004Adh cells (0.42 vs. 0.41) (Figure 2E). 

Next, we examined the effect of AA or DHA treatment on the fatty acid composition of A4-004N and A4-004Adh total lipids (Appendix A). Cells were cultured in their respective “neurosphere” versus “adherent” media, supplemented with BSA (control), 30 μM AA or 30 μM DHA for 24 h before they were harvested for lipid extraction. To ensure that BSA itself would not affect our results, we also compared the fatty acid composition of A4-004N and A4-004Adh lipids in BSA (vehicle control) and untreated cells. No significant differences were noted between untreated and BSA-treated cells (Appendix A). 

AA treatment increased AA levels, along with its downstream mediators, adrenic acid (ADA, C22:4ω-6), and docosapentaenoic acid (DPA, C22:5ω-6) in both A4-004N and A4-004Adh cells (Appendix A). The DHA:AA ratio was decreased ~7-fold and 3-fold upon AA treatment in A4-004N and A4-004Adh, respectively (Figure 3C), whereas the ω-3:ω-6 ratio was decreased ~3-fold and 2-fold upon AA treatment in A4-004N and A4-004Adh, respectively (Figure 3D).

Strikingly, DHA treatment increased DHA content by more than 100-fold (an increase from 0.1% of fatty acids to 11.2% of fatty acids) in A4-004N total lipids compared to 8-fold (1.01% to 8.09%) in A4-004Adh (Figure 3A). This increase in DHA resulted in an ~130-fold increase in the DHA:AA ratio in A4-004N (from 0.07 to 9.65) compared to ~9-fold in A4-004Adh cells (Figure 3C). The overall increase in the ω-3:ω-6 ratio was ~6-fold and ~4-fold in A4-004N and A4-004Adh DHA-treated cells, respectively (Figure 3D). These results indicate efficient uptake of both AA and DHA in neural stem cell-like cultures. 

### 3.2. Fatty Acid Profiles of Phospholipids from GBM Neural Stem-like versus Adherent Cells

Phospholipids are a major component of cell membranes. PUFA supplementation affects membrane phospholipid composition and fatty acid recycling [54,55]. In turn, PUFAs in cell membranes affect their properties, including fluidity and distribution of membrane-bound proteins [49,56,57,58]. We therefore examined the fatty acid composition of phospholipids from A4-004N and A4-004Adh cells (Appendix A and Figure 2). Trends similar to those observed in total lipids were seen in phospholipids, such as increased MUFAs (42.8% vs. 39.2%) and decreased PUFAs (9.9% vs. 14.4%) in A4-004N compared to A4-004Adh cells (Appendix A and Figure 2A). There was less ω-6 PUFA incorporated into the phospholipids from A4-004N (7.1%) compared to A4-004Adh cells (9.9%). Similarly, less ω-3 PUFAs were incorporated in the phospholipids from A4-004N (2.9%) compared to A4-004Adh cells (4.6%) (Figure 2B). Levels of ω-3 PUFAs were similar in total lipids and phospholipids of both A4-004N and A4-004Adh cells, whereas ω-6 PUFAs were more abundant in phospholipids compared to total lipids in A4-004Adh cells, suggesting preferential incorporation of ω-6 PUFAs in membrane phospholipids of adherent cultures (Figure 2B). 

Similar to that observed for total lipids, AA was the most abundant ω-6 PUFA in A4-004Adh phospholipids (3.8%), whereas C20:2ω-6 was the most abundant ω-6 PUFA in A4-004N total phospholipids (3.9%) (Appendix A, Figure 2D). Similar to total lipids, ALA (1.6%) was also abundant in A4-004N phospholipids (Appendix A). Increased levels of DHA in both A4-004N and A4-004Adh total phospholipids (0.2% and 1.8%, respectively) compared to total lipids (undetectable and 1%, respectively) were noted, suggesting preferential incorporation of DHA in phospholipids when cells are grown in their standard culture media (Figure 2C). Similar to total lipids, the DHA:AA ratio was lower in A4-004N compared to A4-004Adh phospholipids (0.2 vs. 0.5). The overall ω-3:ω-6 ratio in A4-004N phospholipids was similar to that observed in A4-004Adh (0.4 vs.0.5) (Appendix A and Figure 2E). 

AA treatment resulted in ~6X and ~2X increases in the AA content of phospholipids extracted from A4-004N and A4-004Adh cells, respectively, whereas DHA treatment increased the DHA content of phospholipids by ~18X and ~3.8X in A4-004N and A4-004Adh cells, respectively (Appendix A). Similar to what was observed in total lipids, AA treatment resulted in ~14-fold and ~5-fold increases in the levels of the AA downstream product, ADA, in A4-004N and A4-004Adh phospholipids, respectively. In DHA-treated cells, the DHA:AA ratio increased by ~21-fold in A4-004N phospholipids, resulting in DHA:AA ratio of 4.9 (Figure 3G). In comparison, there was a ~3.8-fold increase in the ratio of DHA:AA in the phospholipids of DHA-treated A4-004Adh cells compared to BSA control cells, resulting in a DHA:AA ratio of 1.8 (Figure 3G). The overall increase in the phospholipid ω-3:ω-6 ratio was ~3-fold and ~2-fold in A4-004N and A4-004Adh DHA-treated cells, respectively (Figure 3H). Thus, while highly significant, the incorporation of DHA into A4-004N and A4-004Adh phospholipids was not as strikingly different as that observed for total lipids, suggesting that DHA levels may be relatively more stable in phospholipids compared to total lipids in GBM cells. 

### 3.3. FABP7 Facilitates DHA Uptake in GBM Neural Stem-like Cells

FABPs play an important role in lipid-mediated biological processes through the regulation of fatty acid uptake, storage, and trafficking fatty acids to different locations in the cell [59,60]. PUFAs, especially DHA and AA, are preferred ligands for FABP7, with ω-3 DHA having a 4-fold higher binding affinity for FABP7 compared to ω-6 AA based on Isothermal Titration Calorimetry and Lipidex 1000 [41,61]. 

To gain insight into the role of FABP7 in the uptake of ω-3 DHA or ω-6 AA in GBM neural stem-like cells, we investigated the effect of FABP7 knockdown on DHA and AA incorporation in the total lipids and phospholipids of GBM neural stem-like cells. FABP7 was knocked-down in A4-004N cells using lentiviral shRNA constructs (shFABP7-1 and shFABP7-2) (Figure 4). In A4-004N shControl cells, AA and DHA treatment increased the intracellular levels of AA and DHA, respectively, in both total lipid and total phospholipid fractions, as described earlier for non-transfected A4-004N cells (Figure 5). Upon FABP7 depletion, however, DHA uptake into total lipids was significantly reduced in DHA-treated A4-004N cells (6.4% in A4-004 shFABP7-1 cells and 6.6% A4-004N shFABP7-2 cells, compared to 12.9% in A4-004N shControl cells) (Figure 5A). Reduced DHA incorporation also resulted in a decreased DHA:AA ratio and ω-3:ω-6 ratio in total lipids of FABP7-depleted A4-004N cells compared to A4-004N shControl cells (Figure 5C,D). Notably, reduced DHA incorporation was not observed in total phospholipids of FABP7-depleted cells (Figure 5E), nor were the DHA:AA and ω-3:ω-6 ratios affected by FABP7 depletion (Figure 5G,H). Interestingly, even though AA is also a ligand for FABP7, FABP7 depletion had no significant effect on the uptake of AA in either total lipids (Figure 5B) or total phospholipids (Figure 5F). In agreement with these data, neither the DHA:AA ratio nor the ω-3:ω-6 ratio was affected by FABP7 depletion in AA-treated A4-004N cells (Figure 5C,D,G,H). Our combined FABP7 results suggest the presence of a compensatory mechanism for AA uptake in GBM neural stem-like cells when FABP7 is depleted, and a specialized role for FABP7 in the uptake of DHA in total lipids of A4-004N cells. 

FABPs are responsible for both the uptake and trafficking of their fatty acid ligands within cells. FABP7 has previously been shown to increase the formation of lipid droplets, an organelle responsible for lipid storage [62,63]. As our data indicate that FABP7 expression increases the uptake of DHA into total lipids, but not phospholipids, we examined whether FABP7 delivers DHA to lipid droplets. We observed an increased accumulation of lipid droplets when A4-004N shControl cells were cultured in medium supplemented with 30 µM DHA (Figure 6). Intriguingly, upon FABP7 depletion, lipid droplet accumulation was significantly reduced, suggesting that DHA may be preferentially stored in lipid droplets in FABP7-expressing cells (Figure 6A). Quantitative analyses revealed a >2-fold decrease in the average intensity of Nile Red staining per cell upon FABP7 depletion in DHA-treated A4-004N cells (Figure 6B). Similar results were observed in U251 cells and ED501N cells, both of which also express FABP7 (Appendix A) [49]. 

### 3.4. DHA-Mediated Inhibition of GBM Neural Stem-like Cell Migration Is Dependent on FABP7 Expression

We have previously reported that AA promotes, whereas DHA inhibits, U87 GBM cell migration in an FABP7-dependent manner [34]. We therefore used the Transwell migration assay to examine the effect of DHA and FABP7 expression on GBM neural stem-like cell migration. A4-004N shControl cells and A4-004N shFABP7 cells were cultured in neurosphere medium supplemented with 30 μM DHA or BSA control for 24 h before carrying out the Transwell assay. As previously shown for adherent GBM cells, migration rates of FABP7-depleted A4-004N cells were significantly reduced compared to A4-004N shControl cells (Figure 7A,B). DHA treatment resulted in a >60% decrease in the migration of A4-004N shControl cells compared to BSA control (*p* < 0.05). In contrast, the migration of A4-004N shFABP7 cells was not affected by DHA treatment. Our combined results indicate that although DHA uptake is observed in both shControl and shFABP7 cells, it is only when FABP7 is present that uptake of DHA leads to the inhibition of GBM neural stem-like cell migration.

## 4. Discussion

GBM tumors share key features with the development of the central nervous system (CNS) [64]. First, GBM tumors have subpopulations of neural stem-like cells that express neural stem cell markers, such as Nestin, CD133, and FABP7. Second, migration and infiltration of GBM cells share morphologically and structurally similar features to those associated with long-distance migration during neurogenesis in developing brain. Third, the DHA:AA ratio in GBM tissue is similar to that seen in the fetal brain when neural cells are undergoing long distance migrations [30,31]. Notably, FABP7 is essential for both the maintenance of radial glial progenitor cells [65] and neuronal cell migration during brain development [38,39]. In standard GBM adherent cell cultures, overexpression of FABP7 increases, whereas knockdown of FABP7 decreases GBM cell migration and infiltration [16,42]. Moreover, FABP7 is highly expressed in GBM neural stem-like cells [40] and its elevated levels are associated with poor clinical outcome [62]. In combination, these data suggest that FABP7 may be able to hijack the brain’s normal developmental processes for the maintenance of GBM stemness and tumor infiltration in brain parenchyma. 

The brain is highly enriched in long chain PUFAs, especially FABP7′s preferred ligands, ω-3 DHA and ω-6 AA, which make up 8% and 6% of the dry weight of adult human brain, respectively [29,66]. FABP7 is found in both the cytoplasm and nucleus, as well as at the plasma membrane where it is involved in the uptake of PUFAs [34,49]. FABP7 plays different roles in the cell depending on which ligand it is bound to [34,66,67]. DHA has previously been shown to inhibit the migration of FABP7-expressing GBM cells [34,35], as well as sensitize GBM cells to chemotherapy and radiation in vitro [32,37,68,69]. It is already known that dietary DHA can affect the fatty acid composition of lipids in brain tissue [70,71,72], opening the door to the possibility that GBM patient outcome could be improved by increasing the DHA content in the tumor microenvironment. 

Neural stem-like cells are key players in GBM cellular heterogeneity and therapy resistance [46,73]. Gene expression profiling has revealed significant differences between neural stem-like GBM cells cultured under neurosphere conditions and adherent GBM cells cultured in serum-containing medium, with the neural stem-like cells more closely mirroring the original tumor [47,48,74]. Furthermore, compared to their matched serum-differentiated counterparts, GBM neurosphere cultures are enriched in enzymes involved in the PUFA synthesis cascade such as ELOVL2 and FASD2 [19]. Cyclooxygenase 2 (COX-2), which metabolizes AA to its downstream bioactive metabolites (e.g., prostaglandins), is preferentially activated in GBM neural stem-like cells compared to adherent cells [75]. We have previously shown that COX-2 is upregulated in FABP7-expressing GBM cells cultured in AA-rich medium [34]. Interestingly, when patient-derived GBM tumors are sorted into fast- versus slow-cycling (neural stem-like) cells, the latter not only preferentially express FABP7 but have elevated levels of PUFA metabolism intermediates [62]. Thus, there is an emerging link between stemness, FABP7 expression, and increased PUFA metabolism in GBM. 

PUFA metabolism is gaining recognition as an important contributor to GBM tumorigenic properties. However, little is known about the uptake and trafficking of PUFAs in GBM neural stem-like cells. To date, the effect of DHA on GBM fatty acid composition has only been investigated using established GBM cell lines cultured under adherent (differentiation-promoting) conditions [32]. By comparing the fatty acid composition of patient-derived GBM neural stem-like cells with that of their adherent counterparts, we found that DHA treatment effectively increases the DHA content in both total lipids and total phospholipids of GBM neural stem-like cells, but especially in total lipids. Our results further indicate that FABP7 plays an important role in the efficient uptake of DHA in total lipids but not phospholipids. This preferential link between DHA, FABP7, and total lipids is particularly interesting in light of our recent finding that DHA treatment disrupts FABP7 nanodomains clustered on the surface of GBM neural stem-like cells and promotes FABP7 localization to mitochondria [49]. Our previous work also shows that DHA treatment in FABP7-expressing GBM cells promotes the nuclear localization of FABP7 and induction of PPAR activity [34]. Thus, FABP7’s main role in DHA-supplemented GBM cells may be to facilitate the uptake, intracellular transport, and utilization of DHA for functions that are unrelated or not directly related to phospholipids.

Along with the decrease in DHA uptake in total lipids observed in FABP7-depleted GBM neural stem like cells, we also found that FABP7 depletion decreased the formation of lipid droplets in GBM neural stem-like cells cultured in DHA-supplemented medium. Associations between FABP7 expression and lipid droplet formation have been previously reported. For example, FABP7 expression is associated with elevated numbers of lipid droplets in both astrocytes and GBM cells [62,63]. Furthermore, up-regulation of FABP7 expression in U87 adherent GBM cells cultured under hypoxic conditions was accompanied by increased fatty acid uptake and increased formation of lipid droplets [76]. Long believed to simply serve as storage sites for fats, lipid droplets are now known to be hubs that coordinate a wide range of lipid-related functions ranging from delivery of fatty acids to mitochondria, regulation of membrane dynamics, and timed release of bioactive lipids that regulate inflammation [77,78]. DHA has already been implicated in the remodeling of lipid droplets in microglia, with a demonstrated effect on the inhibition of neuro-inflammation [79]. Based on these combined data, one may therefore postulate that FABP7 expression in GBM neural stem-like cells has the potential of inducing an anti-tumorigenic response when cells are cultured in a DHA-rich microenvironment that promotes the formation of DHA-rich lipid droplets.

Consistent with the idea that DHA has anti-tumorigenic properties in FABP7-expressing GBM neural stem-like cells, we found that DHA inhibited the migration of FABP7-expressing GBM neural stem-like cells but had negligible effects on their FABP7-depleted counterparts. These results are consistent with our previous findings using the established adherent U87 GBM cell line [34] and indicate that the migratory properties of FABP7-expressing GBM cells are dependent on the ratio of DHA:AA in the tumor microenvironment. Together, our findings suggest that infiltrative FABP7-expressing GBM neural stem-like cells will be selectively targeted by DHA treatment. Thus, while expression of FABP7 may promote GBM growth in an AA-rich tumor microenvironment, in a DHA-rich microenvironment, FABP7 may inhibit tumor infiltration as the result of increased DHA uptake and utilization. We thus propose that FABP7 is the Achilles’ heel of GBM neural stem-like cells, with the potential of inhibiting the migration/infiltration of these cells in a DHA-rich microenvironment. DHA-rich diets have already been shown to inhibit breast cancer xenograft tumor growth and metastasis [80,81], delay neuroblastoma cancer progression in immunodeficient mice [82], as well as increase the in vitro and in vivo efficacy of chemotherapy drugs used for the treatment of colon cancer [83,84,85,86]. It will be important to investigate whether DHA supplementation can increase DHA levels in both the GBM microenvironment and GBM tissue, thereby paving the way to improved GBM patient outcome, by inhibiting the infiltration of FABP7-expressing GBM neural stem-like cells into brain parenchyma.

## Figures and Tables

**Figure 1 nutrients-13-02664-f001:**
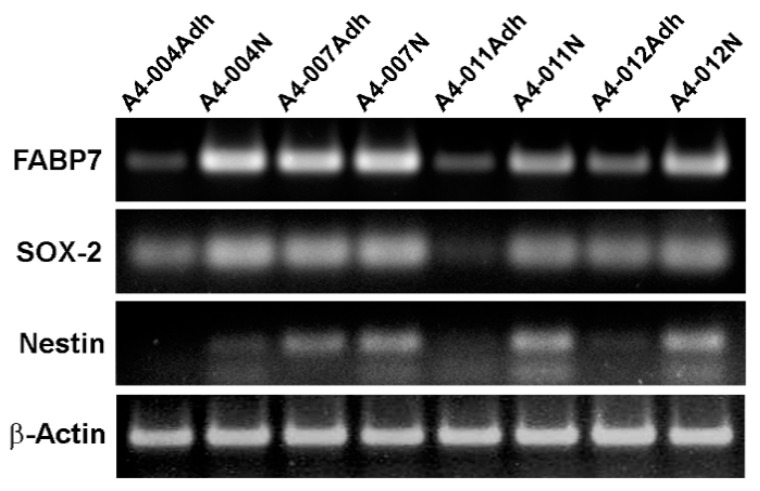
Preferential expression of FABP7 in neurosphere cultures. Semi-quantitative RT-PCR analysis of *FABP7, SOX-2,* and Nestin (*NES*) in paired patient-derived GBM adherent and neurosphere cultures. β-Actin was used as the loading control.

**Figure 2 nutrients-13-02664-f002:**
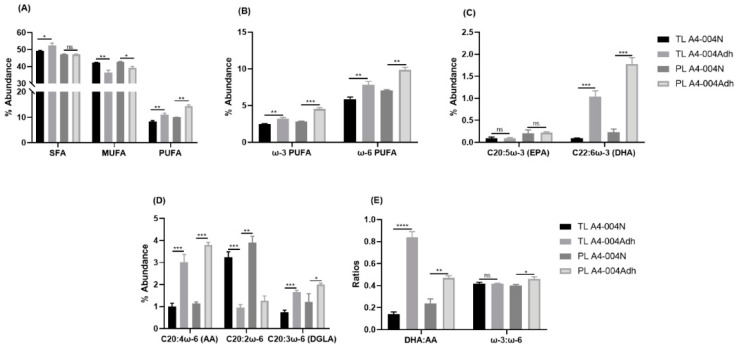
Fatty acid composition of total lipids and total phospholipids extracted from A4-004N and A4-004Adh cells. Total lipids (TL) were extracted, and total phospholipids (PL) separated from A4-004N and A4-004Adh and fatty acid composition measured by gas chromatography. (**A**) Percent abundance of saturated fatty acids (SFA), monounsaturated fatty acids (MUFA), and polyunsaturated fatty acids (PUFA) in total lipids and total phospholipids of A4-004N and A4-004Adh cells. (**B**) Percent abundance of ω-3 PUFA and ω-6 fatty acids PUFA in total lipids and total phospholipids of A4-004N and A4-004Adh cells. (**C**) Percent abundance of C20:5ω-3 (EPA) and C22:6ω-3 (DHA) fatty acids in total lipids and total phospholipids of A4-004N and A4-004Adh cells. (**D**) Percent abundance of C20:4ω-6 (AA), C20:2ω-6 and C20:3ω-6 (DGLA) fatty acids in total lipids and total phospholipids of A4-004N and A4-004Adh cells. (**E**) DHA:AA ratio, and ω-3:ω-6 ratios in total lipids and total phospholipids of A4-004N and A4-004Adh cells. See Appendix A for comprehensive lists of fatty acids in total lipids and phospholipids, respectively. *n* = 3. * indicates *p* < 0.05, ** indicates *p* < 0.01, *** indicates *p* < 0.001, and **** indicates *p* < 0.0001. Abbreviations: DHA, docosahexaenoic acid; AA, arachidonic acid.

**Figure 3 nutrients-13-02664-f003:**
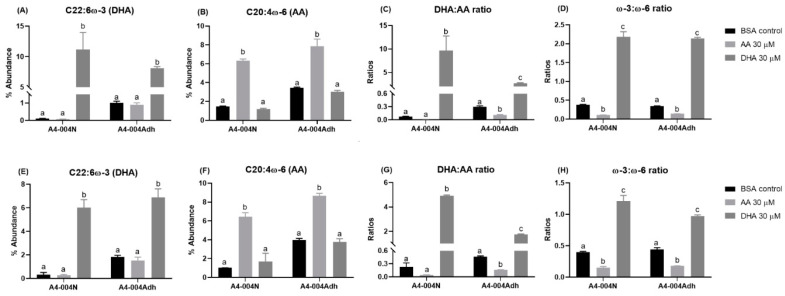
Effects of PUFA treatment on AA- and DHA-incorporation in total lipids and total phospholipids from A4-004N and A4-004Adh cells. A4-004N and A4-004Adh culture media were supplemented with BSA (control), 30 μM AA, or 30 μM DHA. Percent abundance of DHA (**A**,**E**), percent abundance of AA (**B**,**F**), DHA:AA ratio (**C**,**G**) and ω-3:ω-6 ratio (**D**,**H**) in total lipids (**A**–**D**) and total phospholipids (**E**–**H**) of A4-004N and A4-004Adh cells. *n* = 3. Different letters indicate that groups are significantly different. See Appendix A for comprehensive lists of fatty acids in total lipids and phospholipids, respectively.

**Figure 4 nutrients-13-02664-f004:**
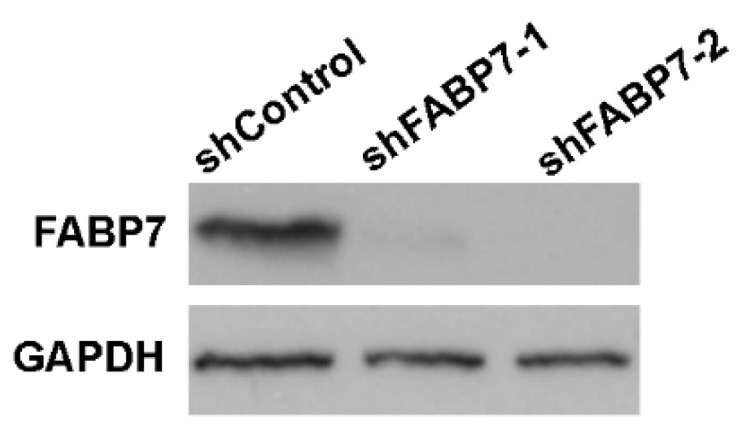
FABP7 levels in lentivirus-infected A4-004N cells. Western blot analysis of FABP7 in total lysates prepared from A4-004N cells transfected with shControl, shFABP7-1 and FABP7-2 lentiviral constructs. GAPDH was used as the loading control.

**Figure 5 nutrients-13-02664-f005:**
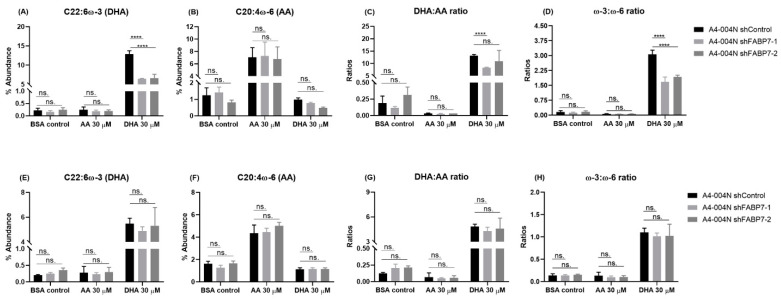
Effects of PUFA treatment on the incorporation of AA and DHA into total lipids and total phospholipids from A4-004N control and shFABP7 knockdown cells. A4-004N shControl, shFABP7-1, and shFABP7-2 were cultured in medium supplemented with BSA (control), 30 μM AA, or 30 μM DHA. Percent abundance of DHA, AA, DHA:AA ratio and ω-3:ω-6 ratio in total lipids (**A**–**D**) and total phospholipids (**E**–**H**) of A4-004N shControl and shFABP7 cells. **** indicates *p* < 0.0001. ns. indicates not significant.

**Figure 6 nutrients-13-02664-f006:**
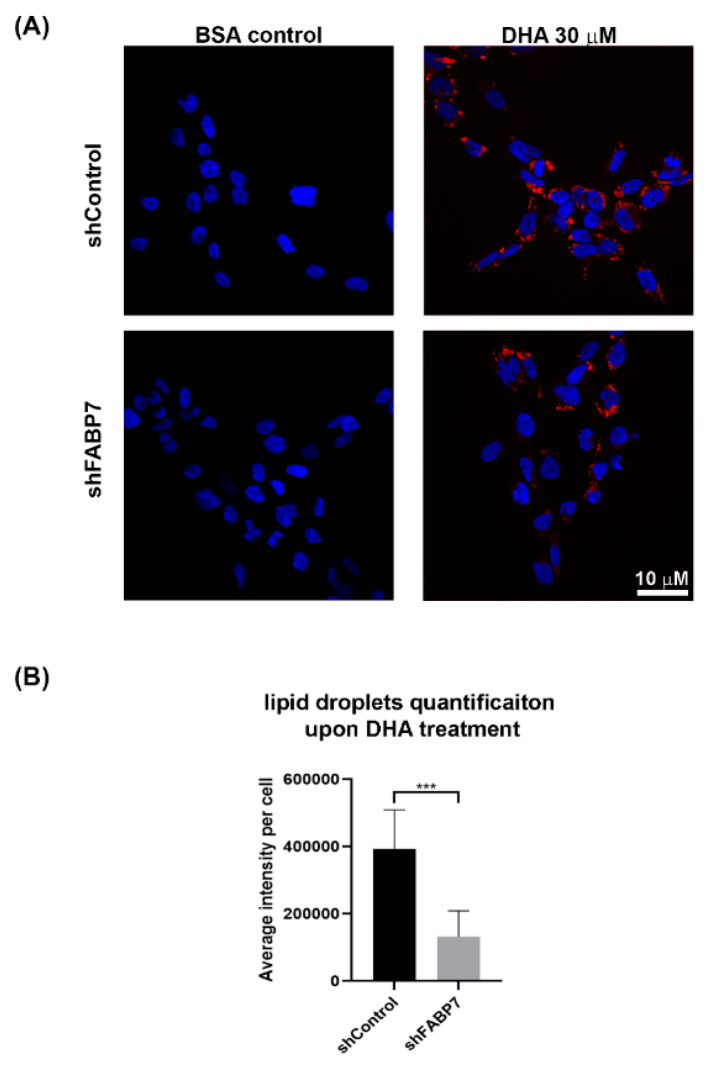
Effect of DHA treatment on lipid droplet formation in A4-004N shControl and A4-004N shFABP7 cells. (**A**) A4-004N shControl and shFABP7 cells were cultured in medium supplemented with BSA control or 30 μM DHA. Cells were stained with Nile Red and images captured by confocal microscopy. DAPI was used to stain the nucleus. (**B**) Quantification of lipid droplets in A4-004N shControl and shFABP7 cells cultured in DHA-supplemented medium. The average intensity of Nile Red staining per cell was measured using raw images (*n* = 8, 15–30 cells/image) taken by confocal microscopy. *** indicates *p* < 0.001.

**Figure 7 nutrients-13-02664-f007:**
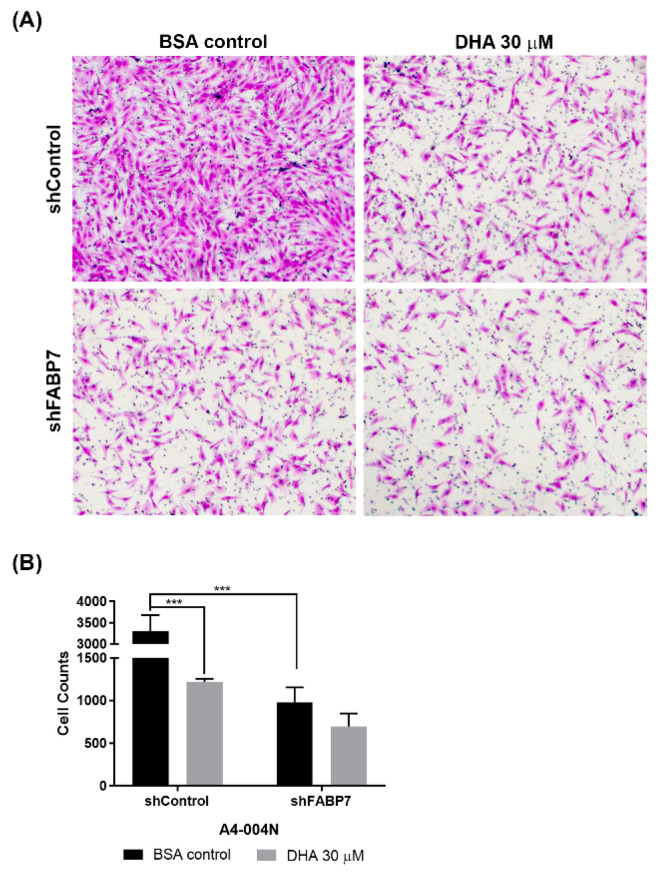
Effects of DHA treatment on the migration of A4-004N shControl and A4-004N shFABP7 cells. (**A**) Representative Transwell images of A4-004N shControl and A4-004N shFABP7 cells cultured in medium supplemented with bovine serum albumin (BSA control) or 30 μM DHA. (**B**) Quantification of migrating cells described in (**A**). Differences were assessed for significance using the two-tailed unpaired *t*-test. *n* = 3. *** indicates *p* < 0.005.

## Data Availability

No new data were created or analyzed in this study. Data sharing is not applicable to this article.

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
