# Peer review of "FABP7 Facilitates Uptake of Docosahexaenoic Acid in Glioblastoma Neural Stem-like Cells"

_nutrients, 2021, doi:10.3390/nu13082664_

Round 1

Reviewer 1 Report

Dear authors,

First of all, I’d like to give a great congratulation to them for nice and successful study. Choi, et al. suggested that increasing DHA content in the GBM microenvironment may reduce the migration/infiltration of FABP7-expressing neural stem-like cancer cells. I think that the topic is interesting enough to be attractive to readers in terms of research on cancer metabolism. Also, their study was well designed, and methods were also reasonable and scientific. Their approach to hypothesis using lipid droplet analysis seems to be much nice.

1) I wonder if authors can show the results of MR spectroscopy findings of real GBM patients. If their results are correct, MR spectroscopic finding can show the same results. The data do not have to be large but had better illustrate several examples of increased DHA in real GBM.

2) In figure 4, they showed the results of Western blotting. However, the explanation was not in the section of material and methods. Please, describe the experiments which they did.

Good luck.

Author Response

Reviewer #1 comments:

  1. I wonder if authors can show the results of MR spectroscopy findings of real GBM patients. If their results are correct, MR spectroscopic finding can show the same results. The data do not have to be large but had better illustrate several examples of increased DHA in real GBM.

Thank you for this suggestion. This would be an interesting study. We hope to be able to do something along these lines in the future. The study would involve a clinical trial with two arms: patients receiving a DHA-supplemented diet and a control group.

  1. In figure 4, they showed the results of Western blotting. However, the explanation was not in the section of material and methods. Please, describe the experiments which they did.

Thank you very much for pointing this out. We completely missed that. We now include details about Western blotting in Materials and Methods (pg. 3, lines 111-118).

Reviewer 2 Report

In this study, the authors address the lipid metabolism in glioblastoma (GBM); in particular, they focus on high expression of the brain fatty acid-binding protein (FABP7) in the GBM neural stem-cell like cells and low level of the polyunsaturated fatty acids (PUFAs) ω-3 docosahexaenoic acid (DHA) in the GBM microenvironment. First, they examined the fatty acid composition in patient-derived GBM neural stem-like cells. Secondly, they investigated the effect of AA and DHA treatment on the fatty acid profiles in GBM neural stem-like cells. Next, they showed that DHA treatment increases DHA levels and the DHA:AA ratio in GBM neural stem-like cells, with FABP7 facilitating DHA uptake. Finally, they found that increased uptake of DHA inhibits migration of GBM neural stem-like cells.

The hypothesis that increasing DHA content in the GBM microenvironment might have the therapeutic value is simple and interesting. Lipid metabolism could become a target mechanism of GBM treatment and their approach could develop a novel therapeutic strategy of GBM.

I have some comments.

  1. Is Title (Page 1, Line 2-3) appropriate for the present study?
  2. In Abstract (Page 1, Line 19-20), “Similar to brain, GBM tissue is enriched in AA and DHA. However, DHA levels are considerably lower in GBM tissue compared to adult brain”. What does that mean? Is DHA “enriched” or “lower levels” in GBM tissue?
  3. In Results (3.1.), the first paragraph (Page 4, Line 168-182) might be too long, and should be included in Introduction or Discussion.
  4. Regarding Figure 1, GBM adherent cultures also express FABP7 (A4-007Adh > A4-012Adh > A4-011Adh), as described in the text. How about the expression level of neural stem cell markers, Nestin and SOX-2, in these adherent cultures? What happens in these adherent cultures, especially in A4-007Adh?
  5. Regarding “the effect of FABP7 knockdown” experiments, how about the results in FABP7-expressing cells (as shown in Figure 1) other than A4-004N?
  6. Regarding “the effect of DHA treatment” experiments, how about the results in GBM neural stem-like cells (as shown in Figure 1) other than A4-004N?
  7. How are the effects of DHA treatment on “stemness” of GBM neural stem-like cells?
  8. Regarding Discussion (Page 12, Line 452-455), DHA-rich diets could result in DHA-rich GBM microenvironment?

Author Response

Reviewer #2 comments:

  1. Is Title (Page 1, Line 2-3) appropriate for the present study?

Although we have data about the effect of DHA on GBM neural stem-like cell migration and lipid droplet formation, the main focus of our paper is on the effect of FABP7 on DHA uptake in GBM neural stem-like cells. Hopefully, the reviewer will agree that the title reflects the work carried out in the manuscript.

  1. In Abstract (Page 1, Line 19-20), “Similar to brain, GBM tissue is enriched in AA and DHA. However, DHA levels are considerably lower in GBM tissue compared to adult brain”. What does that mean? Is DHA “enriched” or “lower levels” in GBM tissue?

Thank you for pointing that out. We have revised the text to make the statements in lines 19-20 clearer.

  1. In Results (3.1.), the first paragraph (Page 4, Line 168-182) might be too long, and should be included in Introduction or Discussion.

The first paragraph of the Results section has been shortened (pg. 4, lines 180-189). The reviewer is right – there was a lot of duplication with the Introduction.  

  1. Regarding Figure 1, GBM adherent cultures also express FABP7 (A4-007Adh > A4-012Adh > A4-011Adh), as described in the text. How about the expression level of neural stem cell markers, Nestin and SOX-2, in these adherent cultures? What happens in these adherent cultures, especially in A4-007Adh?

We now include both Nestin and SOX-2 in Figure 1. As expected, A4-007Adh cells have the highest levels of FABP7, SOX-2, and Nestin RNA compared to the other 3 adherent cultures. We believe that the high levels of stem cell markers in A4-007Adh cells reflects the fact that patient A4-007 glioblastoma tumor has elevated levels of stem-like cells. We predict that the levels of neural stem-like cell markers in A4-007Adh cells will gradually decrease because cells cultured under adherent conditions tend to become less stem cell-like over time.

  1. Regarding “the effect of FABP7 knockdown” experiments, how about the results in FABP7-expressing cells (as shown in Figure 1) other than A4-004N?

  1. Regarding “the effect of DHA treatment” experiments, how about the results in GBM neural stem-like cells (as shown in Figure 1) other than A4-004N?

Please see attached file for A4-007N data

Comment #6: Both comments #5 and #6 are addressed here. We have carried out preliminary gas chromatography analysis on total lipids extracted from A4-007N cells to see whether AA and DHA supplementation changes the lipid composition of A4-007N cells. The results are very similar to that described for A4-004N in our manuscript. DHA and AA supplementation increased DHA and AA levels, respectively, in A4-007N, as well as altered the DHA:AA ratio and the ω-3:ω-6 ratio (data are shown below). However, as this pilot experiment was only done once, precluding statistical analysis, we did not include these data in the manuscript.

Comment #5: We also attempted to knockdown FABP7 in A4-007N cells, but the cells grew very slowly and the experiment could not proceed.  

  1. How are the effects of DHA treatment on “stemness” of GBM neural stem-like cells?

This is a very interesting question which will be followed up in future studies.

  1. Regarding Discussion (Page 12, lines 452-455), DHA-rich diets could result in DHA-rich GBM microenvironment?

This is a very good point which we have incorporated in the Discussion (pg. 12, lines 461-465). 

Round 2

Reviewer 2 Report

The authors responded to the reviewer's comments.